# RAC1 Activation as a Potential Therapeutic Option in Metastatic Cutaneous Melanoma

**DOI:** 10.3390/biom11111554

**Published:** 2021-10-20

**Authors:** Paula Colón-Bolea, Rocío García-Gómez, Berta Casar

**Affiliations:** 1Instituto de Biomedicina y Biotecnología de Cantabria, Consejo Superior de Investigaciones Científicas—Universidad de Cantabria, 39011 Santander, Spain; colonboleapaula@gmail.com (P.C.-B.); rocio.garciagomez@unican.es (R.G.-G.); 2Centro de Investigación Biomédica en Red de Cáncer (CIBERONC), Instituto de Salud Carlos III, 28029 Madrid, Spain

**Keywords:** RAC1, cutaneous melanoma, invasion, metastasis, therapy resistance

## Abstract

Metastasis is a complex process by which cancer cells escape from the primary tumor to colonize distant organs. RAC1 is a member of the RHO family of small guanosine triphosphatases that plays an important role in cancer migration, invasion, angiogenesis and metastasis. RAC1 activation has been related to most cancers, such as cutaneous melanoma, breast, lung, and pancreatic cancer. RAC1P29S driver mutation appears in a significant number of cutaneous melanoma cases. Likewise, RAC1 is overexpressed or hyperactivated via signaling through oncogenic cell surface receptors. Thus, targeting RAC1 represents a promising strategy for cutaneous melanoma therapy, as well as for inhibition of other signaling activation that promotes resistance to targeted therapies. In this review, we focus on the role of RAC1 in metastatic cutaneous melanoma emphasizing the anti-metastatic potential of RAC1- targeting drugs.

## 1. Introduction

Melanoma is the most aggressive form of skin cancer representing more than 80% of deaths in cutaneous malignancies [1]. Metastasis is a complex process involving several steps, including migration, invasion, EMT (Epithelial-Mesenchymal Transition), angiogenesis, survival, plasticity and colonization of secondary tissues [2]. Metastatic cutaneous melanoma is a lethal disease with low survival rates, due to rapid acquisition of resistance to most available therapies [3].

Rho GTPases are important molecules that regulate cellular functions such as growth, motility, survival, migration, invasion, and metastasis thereby affecting tumor progression [4,5,6,7]. Rho GTPases are molecular switches that cycle between an active GTP and an inactive GDP bound form. RAC1 is one of the best characterized Rho GTPases that regulate crucial processes for melanoma tumorigenesis and metastasis. RAC1 activation is tightly regulated by activators, including guanine-nucleotide exchange factors (GEFs), and inhibitors, GTPase-activating proteins (GAPs) and guanine-nucleotide disassociation inhibitors (GDIs). In addition, RAC1 is regulated by modifications at its C terminus, including palmitoylation, carboxymethylation, and geranylgeranylation, as well as numerous post-translational modifications that influence its localization, activity, and ability to bind its effectors. RAC1 expression and activity are increased in human malignancies which in some cases correlates with aggressiveness and poor prognosis.

RAC1 signaling pathway is hyperactivated in human cancers and promotes tumor initiation, progression, and metastatic dissemination [8]. RAC1 point mutations and deregulated stability or subcellular localization have been identified as mechanisms that contribute to tumorigenesis and metastasis. Indeed, The Cancer Genome Atlas (TCGA) data shows that RAC1 is upregulated or mutated in over 10% of human cancers, including melanoma, glioblastoma, skin, esophageal, gastric, bladder, head and neck, liver, pancreatic, and prostate carcinomas [9,10]. RAC1P29S is the third most commonly mutated codon in human cutaneous melanoma, after BRAF V600 and NRAS Q61, and one of the most prominent driver mutations in RAC1 with a frequency of approximately 5% and up to 10% in chronically sun-exposed melanomas [11].

RAC1P29S mutation leads to gain of function and enhances binding to target proteins PAK1 and MAP3K11 (MLK3) activating downstream RAC1 signaling. RAC1P29S has a particular role in early transformation, enhancing cell migration and proliferation [12]. Moreover, this mutation confers resistance to RAF and MEK inhibitors, thus having significance in clinical therapeutic strategies [13]. Interestingly, expression of RAC1P29S in melanoma patients correlates with PD-L1 upregulation contributing to evading antitumor immune response and potentially serving as a predictive biomarker for therapy resistance in melanoma [14].

RAC1 has been implicated in RAS-induced neoplastic transformation. Moreover, malignant melanocytes have elevated RAC1 activity during migration, invasion and metastasis. It has been described that deregulation of GEFs, such as Dbl, Vav, Trio, Ect2, Tiam-1 and P-REX-1 also contribute to aberrant RAC1 signaling in many types of tumors [15,16,17].

In addition, RAC1 signaling can modulate cell motility and invasion through a variety of mechanisms such as promoting membrane protrusions and regulating focal adhesions. RAC1 activation has also been shown to regulate the mode of cell movement to promote colonization of tumor cells. Efficient regulation of RAC1 signaling may be required for cell –cell adhesion, tumor cell migration and invasion during metastasis. Inhibition of RAC1 activity could represent an opportunity to develop novel therapeutic approach to target different stages of tumor cell metastasis [18].

During tumor progression, a significant remodeling of the extracellular matrix (ECM) is evident [19]. RAC1 is involved in invadopodia-mediated ECM degradation [20]. RAC1 activation drives motility by regulating lamellipodia formation, focal adhesions and MMP expression [21].

The best-understood effectors for RAC1 are the p21-activated protein kinases (PAKs). PAKs regulate a multitude of cellular processes including cell motility, survival, proliferation, and organization of the cytoskeleton [22]. PAK1 is overexpressed in a subset of BRAF wildtype melanomas. RAC1 mutant human melanoma cells are resistant to clinical inhibitors of BRAF but are uniquely sensitive to PAK inhibitors [23].

RAC1 can be shuttled from the cytoplasm to the nucleus and abnormal localization, particularly in the nucleus, has been detected in cancer cells. In the nucleus, RAC1 can induce nuclear alterations through nuclear actin promoting nuclear plasticity during invasiveness [24]. Moreover RAC1 localization and activity can be regulated by scaffolding mechanisms. Temporal and spatial localization of RAC1 is tightly regulated by protein-protein interactions [25,26].

RAC1 signaling is also involved in angiogenesis and required for vertical blood vessel sprouting associated with tumor-induced angiogenesis [27]. RAC1 plays an important role in the development of resistance to anti-VEGF therapy, suggesting that the combination of VEGF/VEGFR-targeted therapies with a RAC1 inhibitor may improve the efficacy of the anti-metastasic therapies [28].

In this review, we focus on the role of RAC1 in cutaneous melanoma metastasis, advances in our understanding of key signaling pathways altered by activated RAC1, and its potential clinical therapeutic implications in metastatic cutaneous melanoma treatment. RAC1 hyperactivation plays an important role in regulating resistance to targeted therapies, as well as in the suppression of antitumor immune response, and this highlights the critical need to develop new therapeutic strategies to target RAC1 signaling (Figure 1).

## 2. RAC1 Pathway Activation in Melanoma Formation

There is a growing body of evidence indicating that an enhanced activation of RAC1, either through its overexpression, its hyperactivation by GEFs or the appearance of the P29S mutation, contributes to cutaneous melanoma formation, often associated with activating mutations in *BRAF* or *NRAS*, or inactivating mutation of *NF1* [13,14]. Although the exact downstream pathways by which RAC1 exerts its effects are still being unravelled, there are multiple studies pointing to possible mechanisms.

One of the possible signaling pathways of RAC1 in the promotion of melanoma is through its binding to Bcl-2. It has been shown that Bcl-2 phosphorylated at serine-70 (S70pBcl-2) confers apoptosis resistance to cancer cells [29]. Chong et al. described how RAC1-GTP binds to Bcl-2 leading to the accumulation of S70pBcl-2. Overexpression of RAC1 in melanoma cells increased ROS levels that inhibited PP2A preventing thereby dephosphorylation of Bcl-2. The authors describe a positive feedforward loop between RAC1-GTP and S70pBcl-2 sustaining an anti-apoptotic signaling in these cells [30].

In addition, RAC1P29S has been reported to increase the expression of PD-L1, not only with exogenous expression of RAC1P29S in vitro, but also with endogenous expression in melanoma patients. The exact mechanism by which the oncogene increases the levels of PD-L1 is still unknown, but the authors postulate that this increment helps melanomas to evade the immune system and thereby facilitates its growth [14].

A direct target protein of RAC1 is the lipid kinase phosphatidyl inositol-3 kinase (PI3K)-β [31,32]. Indeed, treatment with selective inhibitors for PI3Kβ in melanoma cell lines harboring the RAC1P29S mutation showed a decrease in proliferation [33]. Other PI3K selective inhibitors, including PI3Kα, δ and γ, appeared to be less effective. Another study relating AKT signaling to survival described how melanocytes with endogenous RAC1P29S had a higher survival rate when cultured in the absence of growth factors or in an anchorage-independent condition. With the use of siRNA and small molecules, the authors were able to associate these events to AKT signaling [34].

RAC1P29S can induce ERK phosphorylation in melanocytes [9]. This capacity of activating MAPK pathway also plays a role in protecting melanoma cells with *RAC1* and *BRAF* mutations from apoptosis when treated with RAF inhibitors. The authors claim that RAC1P29S sustained the levels of pMEK and pERK in the presence of inhibitors and that these levels decreased following RAC1P29S knockdown [13]. In line with the role of RAC1 in this MAPK pathway are the PAKs, one of the best characterized effectors for RAC1 [22]. These proteins promote, among other processes, cell survival and proliferation by phosphorylating different substrates. PAKs have been described as key components of the ERK pathway, not only due to their kinase activity (they are able to phosphorylate c-RAF at S338 and MEK1 at S298), but also to their scaffolding function [35]. Inhibiting PAKs function with Frax-1036 in melanoma cells harboring RAC1P29S mutation, resulted in marked reduction of proliferation and viability. These results were corroborated in in vivo xenograft experiments [23]. Lionarons et al. also described decreased proliferation after genetically inhibiting PAK in their animal model [34]. These results point to a RAC1-PAK- MEK-ERK pathway in the formation of melanoma.

In this last study, the authors also identified another signaling pathway related to the promotion of survival in melanocytes and melanoma cells, independent of PAK or AKT. RAC1P29S activates a WAVE-ARP2/3-SRF/MRTF cascade that triggers a transcriptional program switching the cells to a mesenchymal phenotype characterized by resistance to apoptosis.

## 3. RAC1 Signaling in Tumor Cell Migration and Invasion

Cell migration is required for many processes such as embryogenesis or wound healing, but when deregulated it contributes to dissemination of cancer metastases. Melanoma cells can invade not only the dermis, but also other organs such as the lungs, the liver or the brain [36]. RAC1, as a pivotal regulator of the cytoskeleton, plays a main role in this process. It drives motility by promoting among others, lamellipodia formation, focal adhesions and MMP expression [37].

Tumor cells can switch between two different modes of movement. RAC1 is responsible for directing mesenchymal movement, characterized by an elongated cellular morphology and the requirement of extracellular proteolysis. Sanz-Moreno et al. discovered that when activated, RAC1 suppressed RHOA dependent amoeboid movement through decreasing actomyosin contractility. An important RAC1 effector is WAVE2, a member of the WASP-family verprolin-homologous proteins. These proteins regulate the actin cytoskeleton and therefore have an important role in cell migration and invasion. The authors were able to trace decrease in phosphorylation of Myosin Light Chain (MLC) to WAVE2 [38]. Another study pointing to the importance of WAVE2 as a RAC1 effector, described how in mouse melanoma cells with ectopic overexpression of constitutive active RAC1, there was an increase in invasiveness that was reverted by WAVE2 RNAi [39]. Regulation of the cytoskeleton by WAVEs takes place through the activation of the actin nucleation complex ARP2/3 [40] which in turn leads to the activation of SRF/MRTF inducing a transcription program in melanocytes that leads to mesenchymal transition [34].

Another pathway regulated by RAC1 that plays an important role in melanoma is PI3K-AKT, which has also been described to play a role in EMT [41] and migration [42,43]. PI3K -RAC1 activation regulates EMT in melanoma cells and promotes metastasis.

Regarding MMP expression, there is a study that relates MMP-2 to RAC1 in melanoma cells. The authors saw how P-REX1 (PIP3-dependent RAC exchange factor-1), a guanine nucleotide exchange factor that activates RAC1, regulated cell migration and invasion. Cells bearing overexpression of P-REX1 had increased RAC1-GTP, p-PAK1, p-p38 and MMP-2 levels. To further confirm their results, they used RAC1 and p38 inhibitors in control cells and in P-REX1 knockdown cells. Control cells exhibited a pronounced inhibition in migration and invasion, whereas P-REX1 knockdown cells showed no changes. These results highlight the importance of the P-REX1/RAC1/PAK1/p38/MMP-2 pathway in migration and invasion of melanoma cells [43].

RAC1 regulation of invadopodia in melanoma is controversial. Revach et al. observed how expression of wildtype RAC1 in cells led to invadopodia formation, whereas RAC1P29S harbouring cells, with higher migration rate, showed suppressed invadopodia and matrix degradation, but enhanced lamellipodia formation. These confusing results point to different signaling pathways for wildtype and mutant RAC1. Disassembly of invadopodia has been related to a TRIOGEF-RAC1-PAK1-cortactin pathway [44]. Increased number of lamellipodia in melanoma cells expressing mutant RAC1 compared to wildtype has also been described by Mohan et al. [45]. In this study the authors describe how RAC1P29S induced an enhanced lamellipodial branched actin network conferring the cells higher migration and the ability to sequester and phosphoinactivate Merlin, a tumor suppressor known to downregulate cyclin D1 and prevent cell cycle progression [46]. Interestingly, the authors demonstrate how for the inactivation of Merlin, both PAK activation and the branched actin polymerization driven by mutated RAC1, are necessary. RAC1 through PAK1, Merlin and the cytoskeleton renders the melanoma cells a higher metastatic potential and higher proliferation rate of metastatic cells.

RAC1 not only exerts its pro-metastatic effect through the actin cytoskeleton, but also through its ability to act on the microtubule cytoskeleton. We showed how through PAK1, constitutive active RAC1 acted on the microtubules, and through the Linker of Nucleoskeleton and Cytoskeleton (LINC) complex, connected to the nucleoskleton and induced nuclear plasticity. This allowed the cells to pass through smaller pores and promoted a more invasive phenotype. Disrupting the LINC complex prevented melanoma cells from undergoing invasion [47].

## 4. RAC1 Signaling in Angiogenesis

Angiogenesis, the process by which the formation of new blood vessels arising from pre-existing vasculature occurs, is essential for tumour growth and dissemination [48]. This neovascularization is crucial to maintain oxygen and nutrients supply, and it might serve tumoral cells as a path to colonize distant organs.

Capillary formation is tightly controlled by a pro- and anti- angiogenic factors balance, yet this changes during tumour progression, where the scale tilts towards a pro-angiogenic outcome (angiogenic switch) [49], which ultimately regulates endothelial cells proliferation, survival and migration [50]. Some molecules have been highlighted as pro-angiogenic factors during tumour progression, such as RAC1, MMPs, TIMP, and NCK1 [51].

Although other Rho family members are also involved in cancer-related angiogenesis, RAC1 has been proven to be essential in this process, since its activation controls endothelial cell morphogenesis and motility to form a lumen [52,53]. Accordingly, RAC1 endothelial deletion leads to embryonic mortality [54] as a consequence of defects in major vessel formation and absence of small branched vessels, due to cell migration impairment likely mediated by an F-actin dependent mechanism [55].

Moreover, RAC1 is crucial to coordinate endothelial cell–cell adhesion into vessel structures during capillary formation [56]. Hence, with dominant-negative RAC1 mutants, vascular endothelial cells are unable to endure the morphogenic modifications needed for capillary organization, whilst RHO and CDC42 do not affect the same processes [57].

For a blood vessel to form, Matrix Metalloproteases (MMPs) are required to degrade and remodel the vascular basement membrane and ECM, which in turn allows endothelial cell migration and invasion into the surrounding tissue [58]. During this process pro-angiogenic factors such as Vascular Endothelial Growth Factor (VEGF), Transforming Growth Factor-Beta (TGF-β) and others are released, activating a downstream signaling cascade involving RAC1 [59].

Hypoxia enhances RAC1 activity in cancer cells [60], which in turn is required for HIF1 accumulation [61,62,63]. Additionally to being downstream of VEGF, under hypoxic conditions, there is a positive feedback loop where RAC1 can also upregulate VEGF and other angiogenic factors expression, such as Nitric Oxide Synthase (NOS), Platelet-Derived Growth Factor-Beta (PDGF-β) and Ang-2 in a HIF1-dependent manner [64]. Thus, RAC1 overexpression is associated with high levels of VEGF and Vascular Endothelial Growth Factor Receptor (VEGFR) [65]; whereas its downregulation in vascular endothelial cells causes VEGF-mediated tube formation impairment as well as cell migration, invasion and proliferation inhibition in vitro [66].

RAC1 is also able to modulate VEGF by promoting Reactive Oxygen Species (ROS) production via NAD(P)H oxidase in vascular cells [67]. Then, VEGF binds VEGFR2 whose phosphorylation in turn activates downstream signalling molecules such as ERK1/2, Akt and ROS.

Moreover, it has been demonstrated that IQ motif containing GTPase Activating Protein 1 (IQGAP1), a scaffold protein that harbours a RAC1 binding domain, binds directly to VEGFR, mediating ROS-dependent endothelial migration and proliferation [68], while *IQGAP1* knock-out prevents choroidal neovascularization caused by the VEGFR2-RAC1 signalling axis [69].

PAKs have been found to be essential angiogenesis regulators [22], helping in mouse post-stroke recovery [70]. In this regard, the RAC1-PAK1 pathway has been demonstrated to be involved in anti-VEGF (Bevacizumab) and anti-VEGFR (Sunitinib) drug resistance, since its inhibition lessens stem cell properties and overcomes therapy resistance in prostate cancer [28].

Despite being a promising therapeutical target, the usage of anti-angiogenic compounds in cancer clinical trials yielded disappointing results [71]. Targeted therapy against RAC1 and its associated signalling pathways needs to be further investigated as it may prove useful in human diseases involving anomalous vasculature formation and solid tumour treatment.

## 5. RAC1 Targeting Therapies and Therapy Resistance

The identification of RAC1P29S substitution as the third most recurrently observed activating mutation in cutaneous melanoma [72], has opened new therapeutic opportunities. Nevertheless, like other small GTPases, to target RAC1 protein itself turns out challenging. Even so, other therapeutical strategies have been used to treat tumours bearing this signature, such as: preventing RAC1 localization at the plasma membrane, hampering GTP binding, blocking GEF/RAC1 interaction or targeting its effector molecules (Figure 2, Table 1) [10].

### 5.1. Preventing RAC1 Localization

To exert its biological activities RAC1 requires membrane targeting, being this association achieved via carboxyl-terminal lipid modifications. Regarding its plasma membrane translocation, it is mediated by geranylgeranyl transferases type I (GGTI) that trigger RAC1 prenylation [84].

Owing to the importance of post-translational modifications mediating RAC1 subcellular localization and activation, several compounds able to block these lipid modifications have been developed. Among them, GGTI inhibitors have demonstrated promising in vitro and preclinical outcomes [86], exerting anti-tumorigenic effects in human pancreatic and non-small cell lung cancer xenograft mouse models [88,89]. Conversely, in some cell types, it has been shown that GGTI blockade activates RAC1 [90]; this could be a plausible explanation for the inefficacy shown by GGTI2418 in clinical trials [91].

Impairing isoprenoid synthesis by using statins to block HMG-CoA reductase also reduces RAC1 membrane association and activity [85]. In this regard, Simvastatin represses RAC1-dependent MMP-1 production and reduces RAC1 GTP-bound levels [92]. It has been shown that with the mode of action of prenylation-independent statins the observed effects would be due to nuclear RAC1 degradation [93].

Prenylation prepares RAC1 for S-palmitoylation, which is mediated by Palmitoyl Acyltransferases (PATs), enhancing its stability and membrane association. In fact, RAC1 palmitoylation inhibition interferes with its localization and suppresses RAC1-mediated cell migration [86,94]. Several PAT inhibitors have been developed [61,62] and their usage has been shown to increase RAC1 perinuclear localization whilst decrease RAC1 GTP content.

Nevertheless, due to prenylation and palmitoylation, key roles modulating protein localization and activity, it is necessary to further investigate the consequences of RAC1 inhibition.

### 5.2. Hampering Nucleotide Coupling

RAC1 binding to GTP causes its activation, inducing conformational changes that allow attachment of downstream effectors [63]. EHT-1864 is an inhibitor of the Rac family GTPases and blocks activation by direct binding to all RAC isoforms [83]. Critically, it has been shown to block RAC1-mediated transformation [64], although it has also shown off-target effects [83].

### 5.3. Blocking GEF/RAC1 Interactions

Considering GEFs key role coordinating RAC1 signalling and the benefit of inhibiting certain RAC1 functions, blocking GEF-RAC1 interactions represents an interesting therapeutic option.

NSC23766, the first selective RAC1-GEF blocking agent discovered, inhibits RAC1 interaction with TIAM1 and TRIO [79], countering RAC1 tumorigenic effects in several cancer models [5]. Regardless of its effects, NSC23766 potency is not sufficient to use it in clinical applications [95]. This encouraged in silico screening approaches to look for more potent RAC1-GEF inhibitors, leading to the discovery of ZINC08010136 and ZINC07949036 molecules that block RAC1-TIAM1 interaction without affecting RHOA and CDC42 activation [81]. Another inhibitor that was identified, known as “Compound 4”, impeded RAC1 binding to TIAM1, TRIO and VAV2 [77]. This compound has demonstrated to repress cell adhesion and RAC1-mediated cellular events as well as RAC1-PDGFβ-mediated lamellipodia formation [78].

ZINC69391, another virtually screened RAC1-GEF blocking agent, impedes RAC1-TIAM1 binding and efficiently inhibits highly metastatic breast cancer cell proliferation, cycle progression and migration, showing anti-metastatic effects in mouse models. A more potent ZINC69391 analog, 1A-116, that blocks RAC1-REX1 interaction, has shown anti-metastatic effects in breast cancer models [73].

ITX3 is another blocking agent that impedes RAC1-TRIO binding, although it is not a good candidate for clinical designs due to its low potency [80].

EHop-016, discovered by optimization of NSC23766 chemical structure, prevents RAC1-VAV2 binding suppressing RAC1-driven migration of metastatic cancer cells [75]. Nevertheless, these effects could be due to its promiscuity as this compound has also been proven to lessen PAK1 activation and to target CDC42 [76].

Another NSC23766 derivative, MBQ-167, has demonstrated deep growth inhibition of xenografted breast cancer cells [96].

Moreover, RAC1-DOCK1 binding blockade genetically or pharmacologically, by using TBOPP, has demonstrated to cancel RAC1P29S nucleotide exchange and to lessen melanoma and breast cancer matrix invasion [74].

### 5.4. Targeting RAC1 Effectors

Regardless of attempts to target RAC1 activators or RAC1 itself, inhibition of specific RAC1-effector interactions is, to date, the most effective and direct approach for blocking RAC1 outcomes without affecting other downstream signalling pathways.

Among these effectors, the best described druggable RAC1 effectors are the PAKs, which take part in ERK, β-catenin, Aurora A and Merlin activation. In these regards, PAK inhibitors have shown sensibilization of RAC1P29S mutant melanoma cell lines and xenografts [23]. Considering PAK2 key role in cardiac function, its clinical application is questionable; nevertheless, PAK1 inhibitors might be refined for RAC1-driven cancer treatment [97].

Therapeutic potential of RAC1-mediated ROS production inhibition pushed the hunt for RAC1-p67phox interaction inhibitors, such as Phox-I1 that has been shown to inhibit ROS generation in neutrophils [98]. In addition, a RAC1 agonist, called Deacetylepoxydiene (DA-MED), has been developed and its usage induced apoptosis in NSCLC cells due to a huge production of ROS 2 [82].

Activation of SRF/MRTF transcriptional circuit and WAVE2/ARP2/3 actin axis triggered by RAC1P29S, induces a melanocytic to mesenchymal swap and actin filaments organization that, in turn, facilitates cell migration and metastasis. This way, MRTF depletion or use of SRF/MRTF inhibitors (CCG-1423 and CCG-203971) in melanoma cells abolished melanocytic to mesenchymal transition and PLX4720 co-treatment with CCG-257081 overcame tumour growth in mice [34]. However, SRF/MRTF inhibitors mechanism of action is unknown as they have been shown to bind Pirin [99] a transcription factor implicated in melanoma cells senescence, migration and progression [100,101]. Moreover, actin nucleation and/or polymerization inhibition, for example targeting ARP2/3 or formins, could be used in the treatment of RAC1 mutant tumours [45].

Regarding the PI3K network, RAC1 activates AKT by selectively interacting with PI3Kβ [31]. Considering that selective PI3Kβ inhibitors were able to prevent melanoma cell proliferation and migration driven by mutant RAC1 but not by mutant BRAF, whilst PI3Kα inhibitors had the opposite effect [87]; and restricted PI3K inhibitors activity in RAC1P29S melanocytes [34], it would be interesting to further investigate PI3K inhibitors.

Overall, it has been proven the therapeutical advantage of targeting specific RAC1-effector interactions, while the development of more potent inhibitors that selectively target RAC1 signalling cascades is still necessary.

### 5.5. Therapy Resistance

Cancer therapy efficiency has considerably improved thanks to the usage of molecularly targeted drugs. Nonetheless, the main cause of cancer relapse is the appearance of drug resistance which notably lessens treatment efficacy and even drives it to failure.

Evidence from a clinical study of 45 melanoma patients [102] pointed out RAC1P29S status as a vemurafenib and dabrafenib resistance marker that might help to prognosticate patients response to targeted therapy. Moreover, it has been demonstrated that mutant RAC1 can cause ERK phosphorylation in melanocytes [9] and support MAPK signalling in the presence of RAF inhibitors, as overexpression of mutant RAC1P29S in A375 cells raised phosphor-MEK1/2 levels in presence of dabrafenib, thus conferring resistance to MAPK inhibitors in vitro [13,34].

RAC1P29S triggers the PAK, AKT and WAVE-ARP2/3-SRF/MRTF signalling cascade, inducing a switch from a melanocytic to a mesenchymal-like behavior. As a result, melanoma cells acquire improved tumorigenic capacities due to apoptosis suppression and BRAF inhibitor resistance [34]. In the last few years some SRF/MRTF inhibitors have arisen [103,104] and been used in several preclinical models [105]. In this regard, these molecules could be used in the clinic together with BRAF inhibitors to address melanoma resistance [95].

Regarding immune therapy resistance, whereas PD-L1 expression has been demonstrated to be a poor prognostic factor for malignant melanoma [106], Vu et al. [14] found PD-L1 to be upregulated when RAC1P29S was expressed, whereas it was downregulated when RAC1P29S was depleted. Moreover, using melanoma patient samples in TCGA, they identified a positive correlation between PD-L1 expression and RAC1P29S status when compared to wildtype and other RAC1 mutants. These observations suggest that RAC1P29S mutation could be promoting an antitumoral immune response suppression [14].

Therefore, it may be valuable to evaluate clinically RAC1 mutational signature as a predictive biomarker for MEK/RAF inhibitor and anti-PD-1 and PD-L1 therapy resistance in melanoma patients.

Eventually, considering the latest boost in the development of RAS-targeting inhibitors, it isn’t unthinkable that direct inhibitors for mutant RAC1 could arise. Those therapies would attempt avoiding the appearance of drug resistance related to RAC1 mutations, procuring a new molecular weaponry against cancer.

## 6. Conclusions and Perspectives

An overwhelming body of data indicates that RAC1 is involved in tumorigenesis, proliferation, metastatic dissemination as well resistance to targeted therapies. Thus, the identification of RAC1 driver mutations in melanoma opened new therapeutic avenues for metastatic melanoma treatment, highlighting the need for the development of potential drugs to support personalized treatment approaches based on RAC1 inhibition.

Interestingly, the last studies indicated the role of RAC1 activation in oncogenic signaling through some expected effectors, such as the PAKs, and new candidates like SRF/MRTF, which highlights an important link between actin cytoskeleton and oncogenic transcriptional events. These findings suggest new therapeutic strategies to treat cancers driven by this mutation. However, because neither PAK and SRF/MRTF inhibitors are currently in clinical trials, there is still much work to do to translate these findings into the clinic.

In addition, it has been described that there is a correlation between mutant RAC1 and PD-L1 expression, indicating that the potential use of RAC1 inhibitors in combination with anti-PD/PD-L1 antibodies or other agents that facilitate antitumor immune responses, could represent one potential treatment for melanoma. However, additional studies would be required to check the effectiveness of RAC1 inhibitors as a new immunotherapy treatment for melanoma.

Future work will need to determine the role of RAC1 in therapeutic resistance, and the possibility of blocking RAC1 signaling inhibiting specific GEFs, and direct effectors, such as PAK, PI3K, and specific proteins that regulate actin polymerization. Moreover, further studies might be done to develop direct inhibitors of mutant RAC1.

In summary, the overwhelming data argues for the important roles of the RAC1 signaling pathway in every aspect of cancer progression. The aberrant activity of RAC1, RAC-GEFs, and RAC1 effectors in cancer, together with their involvement in metastasis and therapy resistance, emphasize the rich therapeutic opportunities afforded by inhibition of the RAC1 pathway and will translate into benefits for metastatic melanoma patients in a clinical setting.

## Figures and Tables

**Figure 1 biomolecules-11-01554-f001:**
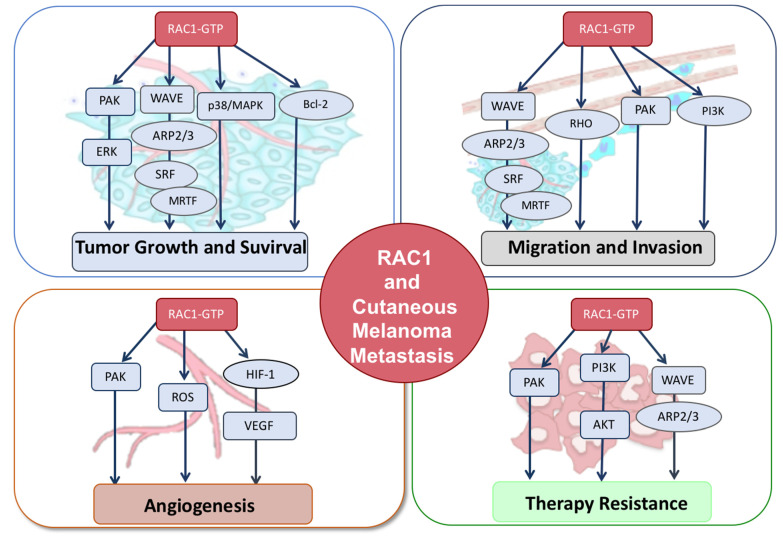
RAC1 signaling in metastatic cutaneous melanoma. Schematic diagram of RAC1 activation regulating tumor growth and survival, migration and invasion, angiogenesis and therapy resistance in cutaneous melanoma.

**Figure 2 biomolecules-11-01554-f002:**
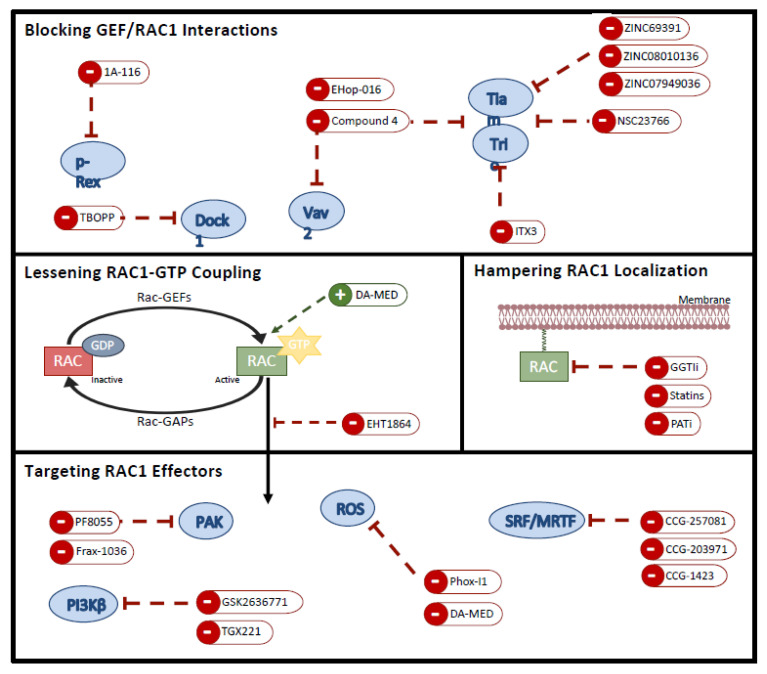
Targeting RAC1 in metastatic cutaneous melanoma. Current available inhibitors to target RAC1 pathway by blocking GEF-RAC1 interactions, lessening nucleotide binding, hampering RAC1 localization as well as by inhibiting downstream effector activity.

**Table 1 biomolecules-11-01554-t001:** Compounds developed targeting RAC1.

Compound Name	Structure	Target	Mechanism of Action	References
Blocking GEF/RAC1 Interactions
1A-116	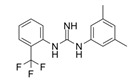	p-REX	Blocks RAC-p-REX1 interaction, reducing intracellular RAC1-GTP levels.	[73]
TBOPP	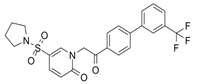	DOCK1	Binds to the DOCK1 DHR-2 domain, inhibiting DOCK1-mediated RAC activation.	[74]
EHop-016	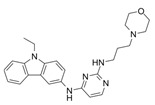	VAV2	Prevents RAC1-VAV2 association, inhibiting activity of RAC downstream effector PAK1. It also targets CDC42.	[75,76]
Compound 4	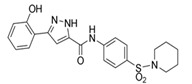	VAV2, TIAM1, TRIO	Impedes RAC1 binding to TIAM1, TRIO and VAV2.	[77,78]
NSC23766	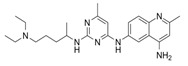	TIAM1, TRIO	Inhibits RAC1 binding and activation by the RAC-specific GEFs TRIO or TIAM1.	[79]
ITX3	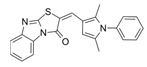	TRIO-N	Inhibits TRIO N-terminal GEF domain, reducing RAC1 and RHOG activation.	[80]
ZINC69391	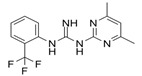	TIAM1	Interferes RAC1-TIAM1 interaction, reducing RAC1-GTP levels.	[73]
ZINC08010136	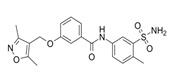	TIAM1	Disrupts RAC1-TIAM1 complex, decreasing active RAC1 cytoplasmic levels without affecting RHOA and CDC42. It is four times more effective than NSC23766.	[81]
ZINC07949036	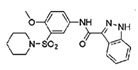	TIAM1	Blocks RAC1-TIAM1 interaction without affecting RHOA and CDC42 activation.	[81]
Lessening RAC1-GTP Coupling
DA-MED	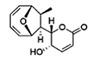	RAC1	RAC1 agonist. It has been shown to induce ROS production.	[82]
EHT-1864	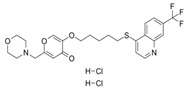	RAC1	Inhibits RAC family GTPases. Blocking its activation by direct binding to RAC1, RAC1b, RAC2 and RAC3.	[83]
Hampering RAC1 Localization
GGTI-2418	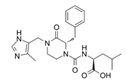	Geranylgeranyl transferase 1	Inhibits GGTase I, in charge of lipid modification required for RAC function.	[84]
Simvastatin	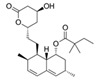	HMG-CoA reductase	Inhibits isoprenoid synthesis, reducing RAC1 membrane association and activity.	[85]
PATi		PATs	Inhibits Palmitoyl Acyltransferases (PATs), interfering with RAC1 localization.	[86]
Targeting RAC1 Effectors
PF8055		PAK	PAK inhibitor	[23]
FRAX1036	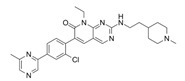	PAK	PAK inhibitor	[25]
GSK2636771	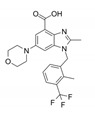	PI3Kβ	Inhibits p110β catalytic subunit of PI3K, impairing AKT phosphorylation by RAC1.	[15]
TGX-221	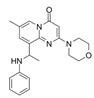	PI3Kβ	Inhibits p110β catalytic subunit of PI3K, impairing AKT phosphorylation by RAC1.	[87]
Phox-I1	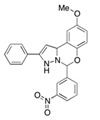	NOX2	Inhibits ROS production in neutrophils, by targeting the p67phox interaction site with RAC1 GTPase.	[87]
CCG-257081	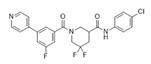	SRF/MRTF	Inhibits SRF/MRTF pathway.	[34]
CCG-203971	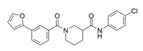	SRF/MRTF	Inhibits SRF/MRTF pathway.	[34]
CCG-1423	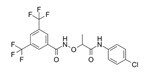	SRF/MRTF	Inhibits SRF/MRTF pathway.	[34]

## Data Availability

Not applicable.

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
