# Peer review of "RAC1 Activation as a Potential Therapeutic Option in Metastatic Cutaneous Melanoma"

_biomolecules, 2021, doi:10.3390/biom11111554_

Round 1
Reviewer 1 Report
In this review, the authors discuss the role of RAC1 in metastatic melanoma.
Whilst metastatic melanoma is indeed a topic worthy of review, there are already other recent reviews on the role of RAC1 in melanoma (for example: "Rac1, A Potential Target for Tumor Therapy" in Front. Oncol., May 2021 and "RAC1 as a Therapeutic Target in Malignant Melanoma" in Trends Cancer., June 2020) which are more comprehensive and better written than this review. Thus, I do not believe that this review will be much cited.
I have strong concerns as to why this this review is not suitable for publication.
(1) In this review, melanoma is treated as a single entity, however, cutaneous melanoma, mucosal melanoma and acral melanoma have distinct underlying genetics and signalling pathways, which are not considered.
(2) The manuscript has not been well written.
(a) It is littered with typographical and grammatical errors. In several places, the sentences do not make sense. I have given some examples below, but there are numerous more:
- Citation of reference [1] is missing. The first reference citations (line 26) are “[2-5]”.
- Line 26: “a n” needs to be changed to “an”.
- Line 27: “GDF” should be “GDP”.
- Line 30: remove the word “including”.
- Line 40: the comma should be removed.
- Line 42: “[7] [8]” should be “[7-8]”.
- Line 44: a comma is needed after “NRAS Q61”.
- Line 44: “RAC wit a requency” should be “RAC1 with a frequency”.
- Line 46: change “P29S” to “RAC1P29S”
- Line 56: “It has een” should be “It has been”.
- Lines 54-57 of the Introduction do not make sense (“Moreover, malignant melanocytes have elevated RAC activity that extended into the epidermis, deeper into the dermis, and was maintained during metastasis”). Similarly, the following lines (57-60) also do not make sense as there are gene names (some given in full, some only in abbreviated form) mixed in with biological processes (“T-cell invasion”).
- Line 73: “can shuttled from de cytoplasm” should be “can be shuttled from the cytoplasm.
- Line 91: “There is a growing number of evidence indicating that” should be “There is a growing body of evidence indicating that”.
- Line 94, 110 and many other places: you cannot start a sentence with “And”.
- Line 100: “Overexpression of RAC1 in melanoma cells increased ROS that inhibited PP2A preventing thereby dephosphorylation of Bcl-2”. This sentence does not make sense.
- Line 125: “its” should be “their”.
- Line 143: “parts” should be “organs”.
- Line 147: “RAC1 is in encharged of….” should be “RAC1 is responsible for…..”.
- Line 150: a comma is needed between “that when activated” and “RAC1 suppressed”.
- Line 151: “An important RAC1 effector is WAVE2, member…” should read “An important RAC1 effector is WAVE2, a member….”.
- Line 154: “MLC” is not defined. Please write in full.
- Line 155: “importance of WAVE2 as RAC1 effector” should be “importance of WAVE2 as a RAC1 effector”.
- Line 157: “invasivenees” should be “invasiveness”.
- Line 161: change to “Another pathway regulated by RAC1 that plays an important role in melanoma is PI3K-AKT, which has also been described to play a role in epithelial-mesenchymal transcription and migration [37] [38].”
(b) The authors are inconsistent with nomenclature, as the manuscript flicks between using “RAC”, “RAC1” and “RAC-1”.
(c) There are numerous places where statements are made but there is no reference cited to back up the statement.
(d) Section 5, subheading "Blocking GEF/RAC1 Interactions": Some paragraphs are only 1 sentence long! This section is not well written and is merely a listing of every paper related to the topic. No assimilation of the findings into a comprehensive report.
(3) It is worth noting that this review cites only 1 reference from 2021 (which in itself is a review!) and only 6 references from 2020. If there has been no more new information in this field, then it begs the questions of why this review is needed?
Author Response
(1) In this review, melanoma is treated as a single entity, however, cutaneous melanoma, mucosal melanoma and acral melanoma have distinct underlying genetics and signalling pathways, which are not considered.
We thank the reviewer for the comment. In this review we have considered cutaneous melanoma. We know that melanoma is not a single entity. The different types of melanoma, cutaneous melanoma, mucosal melanoma and acral melanoma have distinct underlying genetics and signaling pathways. In this review we focus in the role of RAC1 activation in metastasis of cutaneous melanoma, thus we have included the “cutaneous melanoma“ concept in the tittle and other sections.
( 2)The manuscript has not been well written.
We thank the reviewer for the comment. We have corrected all the mistakes and improved the manuscript writing.
(a) It is littered with typographical and grammatical errors. In several places, the sentences do
not make sense. I have given some examples below, but there are numerous more: We thank the reviewer for these suggestions:
• Citation of reference [1] is missing. The first reference citations (line 26) are “[2-5]”. Corrected
• Line 26: “a n” needs to be changed to “an”. Corrected
• Line 27: “GDF” should be “GDP”. Corrected
• Line 30: remove the word “including”. Corrected
• Line 40: the comma should be removed. Corrected
• Line 42: “[7] [8]” should be “[7-8]”. Corrected
• Line 44: a comma is needed after “NRAS Q61”. Corrected
• Line 44: “RAC wit a requency” should be “RAC1 with a frequency”. Corrected
• Line 46: change “P29S” to “RAC1P29S”. Corrected
• Line 56: “It has een” should be “It has been”. Corrected
• Lines 54-57 of the Introduction do not make sense (“Moreover, malignant melanocytes have elevated RAC activity that extended into the epidermis, deeper into the dermis, and was maintained during metastasis”). Similarly, the following lines (57-60) also do not make sense as there are gene names (some given in full, some only in abbreviated form) mixed in with biological processes (“T-cell invasion”). Corrected
• Line 73: “can shuttled from de cytoplasm” should be “can be shuttled from the cytoplasm. Corrected
• Line 91: “There is a growing number of evidence indicating that” should be “There is a growing body of evidence indicating that”. Corrected
• Line 94, 110 and many other places: you cannot start a sentence with “And”.
• Line 100: “Overexpression of RAC1 in melanoma cells increased ROS that inhibited PP2A preventing thereby dephosphorylation of Bcl-2”. This sentence does not make sense. Corrected
• Line 125: “its” should be “their”. Corrected
• Line 143: “parts” should be “organs”. Corrected
• Line 147: “RAC1 is in encharged of….” should be “RAC1 is responsible for…..”. Corrected
• Line 150: a comma is needed between “that when activated” and “RAC1 suppressed”. Corrected
• Line 151: “An important RAC1 effector is WAVE2, member…” should read “An important RAC1 effector is WAVE2, a member….”. Corrected
• Line 154: “MLC” is not defined. Please write in full. Corrected
• Line 155: “importance of WAVE2 as RAC1 effector” should be “importance of WAVE2 as a RAC1 effector”. Corrected
• Line 157: “invasivenees” should be “invasiveness”. Corrected
• Line 161: change to “Another pathway regulated by RAC1 that plays an important role in melanoma is PI3K-AKT, which has also been described to play a role in epithelial-mesenchymal transcription and migration [37] [38].” Corrected
(b) The authors are inconsistent with nomenclature, as the manuscript flicks between using “RAC”, “RAC1” and “RAC-1”. Corrected
(c) There are numerous places where statements are made, but there is no reference cited to back up the statement. Corrected
(d) Section 5, subheading "Blocking GEF/RAC1 Interactions": Some paragraphs are only 1 sentence long! This section is not well written and is merely a listing of every paper related to the topic. No assimilation of the findings into a comprehensive report.
We thank the reviewer for this comment. This section describes GEF-RAC1 interaction as an interesting therapeutic option. We have included a table (Table 1) showing RAC inhibitors that block GEF/RAC1 interactions.
(3) It is worth noting that this review cites only 1 reference from 2021 (which in itself is a review!) and only 6 references from 2020. If there has been no more new information in this field, then it begs the questions of why this review is needed?
We thank the reviewer for this comment. However, we think that our review is a very detailed work, referring to several RAC mutations, signaling pathways, in vivo xenografts, transcriptional programming and therapy resistance. So, we think this review is needed.
Reviewer 2 Report
I really enjoyed reviewing ''RAC1 activation as a potential therapeutic option in metastatic melanoma''. RAC1 is a member of the RHO family of small guanosine phosphatases with great potential in cancer migration, invasion, Angiogenesis and metastasis.
It is a very detailed work, referring to several mutations, signalling pathways, in vivo xenografts, transcriptional programming, therapy resistance.
What I would like to see as a detailed introductory part is the effect of ECM molecules and signalling cascades in melanoma metastasis. Start by describing melanoma stages, existing melanoma therapeutic strategies. You can be inspired by the publications by Stephane Brezillon, where his laboratory is devoted in melanoma metastasis, invadopodia formation, signalling cascades in metastatic melanoma, e.t.c. .
Minor details: where is ref 1 cited in the manuscript ?
line 56 & line 73: check grammar errors
in vivo, in vitro in italics
lines 161- 163 : expand or delete this paragraph
graph 1 is very nicely detailed with reference to tumor growth & survival, migration & invasion, angiogenesis, therapy resistance
graph 2 could have improved analysis
Author Response
We would like to express our sincere appreciation for the reviewers’ constructive comments concerning our review titled “RAC1 activation as a potential therapeutic option in metastatic cutaneous melanoma”(manuscript ID: biomolecules-1385426).
Thanks to their invaluable and professional feedbacks, we believe that the quality of our review has now been significantly improved. We have made extensive modifications to our review according to the reviewers’ comments to finally make our results more comprehensive.
I really enjoyed reviewing ''RAC1 activation as a potential therapeutic option in metastatic melanoma''. RAC1 is a member of the RHO family of small guanosine phosphatases with great potential in cancer migration, invasion, Angiogenesis and metastasis. It is a very detailed work, referring to several mutations, signalling pathways, in vivo xenografts, transcriptional programming, therapy resistance.
We really thank the reviewer for this comment.
What I would like to see as a detailed introductory part is the effect of ECM molecules and signalling cascades in melanoma metastasis. Start by describing melanoma stages, existing melanoma therapeutic strategies. You can be inspired by the publications by Stephane Brezillon, where his laboratory is devoted in melanoma metastasis, invadopodia formation, signalling cascades in metastatic melanoma, e.t.c. ..
We thank the reviewer for the suggestion. We have included the effect of RAC1 activation regulating ECM in melanoma metastasis.
Minor details: where is ref 1 cited in the manuscript ? Thank you for the correction. It was a mistake, we have corrected it.
line 56 & line 73: check grammar errors. We have corrected it.
in vivo, in vitro in italics We have corrected it.
lines 161- 163 : expand or delete this paragraph. We have expanded this paragraph including new data about the role of RAC1 signaling regulating ECM.
graph 1 is very nicely detailed with reference to tumor growth & survival, migration & invasion, angiogenesis, therapy resistance We thank the reviewer for this comment.
graph 2 could have improved analysis We thank the reviewer for this comment.
Reviewer 3 Report
The article “RAC1 activation as a potential therapeutic option in metastatic melanoma” review a growing body of literature on this RHO-family member of small guanosine triphosphatases in malignant melanoma
While the manuscript is generally well written and informative, it may benefit from a more accurate revision of the language in few points; for example:
- Page 4, lines 147-148: “RAC1 is in encharged of ….”
- Page 4, lines 161-163: “epithelial-mesenchymal transcription….”
- Page 5, line 186
- Page 7, lines 285-286:
And others throughout the manuscript.
I believe that the article may also benefit from the inclusion of a table summarizing the compounds developed targeting RAC1 with a description of the mechanism of action and reference to the original paper.
Author Response
We would like to express our sincere appreciation for the reviewers’ constructive comments concerning our review titled “RAC1 activation as a potential therapeutic option in metastatic cutaneous melanoma”(manuscript ID: biomolecules-1385426).
Thanks to their invaluable and professional feedbacks, we believe that the quality of our review has now been significantly improved. We have made extensive modifications to our review according to the reviewers’ comments to finally make our results more comprehensive.
Reviewer 3#
The article “RAC1 activation as a potential therapeutic option in metastatic melanoma” review a growing body of literature on this RHO-family member of small guanosine triphosphatases in malignant melanoma.
While the manuscript is generally well written and informative, it may benefit from a more accurate revision of the language in few points; for example:
We thank the reviewer for these corrections
- Page 4, lines 147-148: “RAC1 is in encharged of ….” Corrected
- Page 4, lines 161-163: “epithelial-mesenchymal transcription….” Corrected
- Page 5, line 186 Corrected
- Page 7, lines 285-286: Corrected
And others throughout the manuscript. We thank the reviewer for this comment. We have corrected the mistakes.
I believe that the article may also benefit from the inclusion of a table summarizing the compounds developed targeting RAC1 with a description of the mechanism of action and reference to the original paper. We thank the reviewer for this suggestion. We have included a table summarizing the compounds developed targeting RAC1 with a description of the mechanism of action and reference to the original paper (Table1).
Round 2
Reviewer 1 Report
Overall the manuscript has been improved, however, there are still some remaining issues:
- Figure 1 still says “RAC” throughout instead of “RAC1”.
- These sentences do not make sense (i.e., I do not understand the point that is being made, so cannot correct them):
Lines 282 – 284
Lines 339 – 340
Lines 353 – 356
Lines 430 – 431
Lines 455 – 457
- The following corrections (“>>>”) need to be made:
Line 15: melanoma >>> cutaneous melanoma
Line 25: cancers >>> cancer
Line 27: define EMT (as this is the first time you use this abbreviation)
Line 69: RAS induced >>> RAS-induced
Line 94, 214, 219 and 484: wt >>> wildtype
Line 97: de >>> the
Line 107: anti-metastasis >>> anti-metastatic
Line 108, 110 and 116: melanoma >>> cutaneous melanoma
Line 120: italicize gene names
Line 123: RAC-1 >>> RAC1
Line 125: renders >>> confers
Line 135: remove “known”
Line 145: RAC1 >>> RAC1 (as it’s a gene name)
Line 146: BRAF >>> BRAF (as it’s a gene name)
Line 206: shRNA >>> knockdown
Line 198: use the abbreviation ‘EMT’
Line 200: promoting >>> and promotes
Line 210: et al –> et al.
Line 211: RAC1 wt >>> wildtype RAC1
Line 212: had >>> showed
Line 215: remove the full stop in the middle of the line
Line 222: is the word “prevent” missing? i.e., “Merlin, a tumor suppressor known to downregulate cyclin D1 and prevent cell cycle progression”
Line 223: both, >>> , both
Line 227 pro metastatic >>> pro-metastatic
Line 233: invasion >>> undergoing invasion
Line 245: pointed out >>> highlighted
Line 245: in >>> during
Line 277: [[53] . >>> [53].
Line 279: major vessels >>>major vessel
Line 280: probably >>> likely
Line 285: Matrix Metalloproteinases >>> matrix metalloproteases
Line 286:remodel >>> remodel the
Line 286: remove “Extracellular Matrix” as ECM has already been defined previously
Line 286: ECM >>> ECM,
Line 287: allow >>> allows
Line 287: cells >>> cell
Line 290: “comprising”….did you mean “involving”?
Line 292: “downstream VEGF”…..did you mean “downstream of VEGF”
Line 298: caused >>> causes
Line 302: did you mean to include “ROS”?
Line 305: VEGFR mediating >>> VEGFR, mediating
Line 306: if you are referring to the IQGAP1 gene then it needs it be italicized (and in lower case if it’s a mouse gene)
Line 306: VEGFR2-RAC1 >>> the VEGFR2-RAC1
Line 309: RAC1-PAK1 >>> the RAC1-PAK1
Line 312: conquers >>> overcomes
Line 316: be used >>> prove useful
Line 320: melanoma >>> cutaneous melanoma
Line 348: remove “and”
Line 350: remove the comma
Line 365: interfere >>> interferes
Line 366: PATs >>> PAT
Line 367: its >>> their
Line 367: shown >>> been shown
Line 367: decreasing >>> decrease
Line 369: palmitoylation >>> palmitoylation,
Line 370: when you say “its inhibition”…..did you mean ‘RAC1 inhibition’? This is not clear
Line 374: change to: “allow attachment of downstream effectors”
Line 374: remove sentence starting “Since RAC1 has high….”. The following sentence should be: “EHT-1864 is an inhibitor of the Rac family GTPases and blocks activation by direct binding to all RAC isoforms [reference]. Critically, it has been shown to block RAC1-mediated transformation [64], although it has also shown off-target effects [reference]”.
Line 378: remove sentence starting “Unluckily, EHT1864 has shown….”
Line 383: depicts >>> represents
Line 402: Change the sentence starting “Then, these molecules…” to read “Another inhibitor that was identified, known as ‘Compound 4’, impeded RAC1 binding to TIAM1, TRIO and VAV2 [87]”
Line 408: cells >>> cell
Line 423: this summary sentence can be removed as it doesn’t contribute anything.
Line 426: Regardless attempts >>> Regardless of attempts
Line 429: druggable RAC1 target PAKs proteins that take part in >>> druggable RAC1 effectors are the PAKs, which take part in….
Line 443: SRF/MRTF inhibitors (CCG-1423 and CCG-203971) usage >>> use of SRF/MRTF inhibitors (CCG-1423 and CCG-203971)
Line 445: overcome >>> overcame
Line 447: progressioN >>> progression
Line 449: could be used in RAC1 mutant tumours treatment >>> could be used in the treatment of RAC1 mutant tumors
Line 451: Regarding PI3K network >>> Regarding the PI3K network
Line 452: cells >>> cell
Line 453: RAC1 and BRAF --- if these are genes (not proteins) they should be italicized
Line 469: RAC1P29S mutant >>> mutant RAC1P29S
Line 470: dabrafenib conferring >>> dabrafenib, thus conferring
Line 472: did you mean “RAC1P29S triggers the PAK, AKT….”
Line 475: inhibitors >>> inhibitor
Line 475: las >>> last
Line 476: and used >>> and been used
Line 480: melanoma [106]. Ha Linh Vu and collaborators, found….. >>> melanoma [106], Vu et al. [14] found…..
Line 487: remove the sentence starting “Further research on aberrant” as it doesn't make sense. Then the following sentence (“Therefore it may be valuable….”) can come straight after the sentence (“These observations suggest that….”). This flows better.
Line 514: RAC1 mutant >>> mutant RAC1
Author Response
We would like to express our sincere appreciation for reviewer 1’ constructive comments concerning our review titled “RAC1 activation as a potential therapeutic option in metastatic cutaneous melanoma”(manuscript ID: biomolecules-1385426).
Thanks to their invaluable and professional feedbacks, we believe that the quality of our review has now significantly improved.
We have made modifications to our review according to reviewer 1’s comments to finally make our results more comprehensive.
Overall the manuscript has been improved, however, there are still some remaining issues:
- Figure 1 still says “RAC” throughout instead of “RAC1”.
We really thank the reviewer for this comment. We corrected it
- These sentences do not make sense (i.e., I do not understand the point that is being made, so cannot correct them):
We really thank the reviewer for this comment. We corrected them.
Lines 282 – 284
Lines 339 – 340
Lines 353 – 356
Lines 430 – 431
Lines 455 – 457
- The following corrections (“>>>”) need to be made:
We thank the reviewer for this comment. Thanks to her/his exhaustive corrections. We corrected all of them. We believe that the quality of our review has now significantly improved.
Line 15: melanoma >>> cutaneous melanoma
Line 25: cancers >>> cancer
Line 27: define EMT (as this is the first time you use this abbreviation)
Line 69: RAS induced >>> RAS-induced
Line 94, 214, 219 and 484: wt >>> wildtype
Line 97: de >>> the
Line 107: anti-metastasis >>> anti-metastatic
Line 108, 110 and 116: melanoma >>> cutaneous melanoma
Line 120: italicize gene names
Line 123: RAC-1 >>> RAC1
Line 125: renders >>> confers
Line 135: remove “known”
Line 145: RAC1 >>> RAC1 (as it’s a gene name)
Line 146: BRAF >>> BRAF (as it’s a gene name)
Line 206: shRNA >>> knockdown
Line 198: use the abbreviation ‘EMT’
Line 200: promoting >>> and promotes
Line 210: et al –> et al.
Line 211: RAC1 wt >>> wildtype RAC1
Line 212: had >>> showed
Line 215: remove the full stop in the middle of the line
Line 222: is the word “prevent” missing? i.e., “Merlin, a tumor suppressor known to downregulate cyclin D1 and prevent cell cycle progression”
Line 223: both, >>> , both
Line 227 pro metastatic >>> pro-metastatic
Line 233: invasion >>> undergoing invasion
Line 245: pointed out >>> highlighted
Line 245: in >>> during
Line 277: [[53] . >>> [53].
Line 279: major vessels >>>major vessel
Line 280: probably >>> likely
Line 285: Matrix Metalloproteinases >>> matrix metalloproteases
Line 286:remodel >>> remodel the
Line 286: remove “Extracellular Matrix” as ECM has already been defined previously
Line 286: ECM >>> ECM,
Line 287: allow >>> allows
Line 287: cells >>> cell
Line 290: “comprising”….did you mean “involving”?
Line 292: “downstream VEGF”…..did you mean “downstream of VEGF”
Line 298: caused >>> causes
Line 302: did you mean to include “ROS”?
Line 305: VEGFR mediating >>> VEGFR, mediating
Line 306: if you are referring to the IQGAP1 gene then it needs it be italicized (and in lower case if it’s a mouse gene)
Line 306: VEGFR2-RAC1 >>> the VEGFR2-RAC1
Line 309: RAC1-PAK1 >>> the RAC1-PAK1
Line 312: conquers >>> overcomes
Line 316: be used >>> prove useful
Line 320: melanoma >>> cutaneous melanoma
Line 348: remove “and”
Line 350: remove the comma
Line 365: interfere >>> interferes
Line 366: PATs >>> PAT
Line 367: its >>> their
Line 367: shown >>> been shown
Line 367: decreasing >>> decrease
Line 369: palmitoylation >>> palmitoylation,
Line 370: when you say “its inhibition”…..did you mean ‘RAC1 inhibition’? This is not clear
Line 374: change to: “allow attachment of downstream effectors”
Line 374: remove sentence starting “Since RAC1 has high….”. The following sentence should be: “EHT-1864 is an inhibitor of the Rac family GTPases and blocks activation by direct binding to all RAC isoforms [reference]. Critically, it has been shown to block RAC1-mediated transformation [64], although it has also shown off-target effects [reference]”.
Line 378: remove sentence starting “Unluckily, EHT1864 has shown….”
Line 383: depicts >>> represents
Line 402: Change the sentence starting “Then, these molecules…” to read “Another inhibitor that was identified, known as ‘Compound 4’, impeded RAC1 binding to TIAM1, TRIO and VAV2 [87]”
Line 408: cells >>> cell
Line 423: this summary sentence can be removed as it doesn’t contribute anything.
Line 426: Regardless attempts >>> Regardless of attempts
Line 429: druggable RAC1 target PAKs proteins that take part in >>> druggable RAC1 effectors are the PAKs, which take part in….
Line 443: SRF/MRTF inhibitors (CCG-1423 and CCG-203971) usage >>> use of SRF/MRTF inhibitors (CCG-1423 and CCG-203971)
Line 445: overcome >>> overcame
Line 447: progressioN >>> progression
Line 449: could be used in RAC1 mutant tumours treatment >>> could be used in the treatment of RAC1 mutant tumors
Line 451: Regarding PI3K network >>> Regarding the PI3K network
Line 452: cells >>> cell
Line 453: RAC1 and BRAF --- if these are genes (not proteins) they should be italicized
Line 469: RAC1P29S mutant >>> mutant RAC1P29S
Line 470: dabrafenib conferring >>> dabrafenib, thus conferring
Line 472: did you mean “RAC1P29S triggers the PAK, AKT….”
Line 475: inhibitors >>> inhibitor
Line 475: las >>> last
Line 476: and used >>> and been used
Line 480: melanoma [106]. Ha Linh Vu and collaborators, found….. >>> melanoma [106], Vu et al. [14] found…..
Line 487: remove the sentence starting “Further research on aberrant” as it doesn't make sense. Then the following sentence (“Therefore it may be valuable….”) can come straight after the sentence (“These observations suggest that….”). This flows better.
Line 514: RAC1 mutant >>> mutant RAC1